# Genistein Promotes M2 Macrophage Polarization via Aryl Hydrocarbon Receptor and Alleviates Intestinal Inflammation in Broilers with Necrotic Enteritis

**DOI:** 10.3390/ijms25126656

**Published:** 2024-06-17

**Authors:** Shuli Quan, Jingxi Huang, Guiqin Chen, Anrong Zhang, Ying Yang, Zhenlong Wu

**Affiliations:** College of Animal Science & Technology, China Agricultural University, Beijing 100193, China; shuliquan123@163.com (S.Q.); huangjx52@163.com (J.H.); guiqin.chen@outlook.com (G.C.); 15504578605@163.com (A.Z.); wuzhenlong@cau.edu.cn (Z.W.)

**Keywords:** genistein, aryl hydrocarbon receptor, necrotic enteritis, M2 macrophages, intestinal inflammation

## Abstract

The aryl hydrocarbon receptor (AhR) is a transcription factor that regulates the immune system through complicated transcriptional programs. Genistein, an AhR ligand, exhibits anti-inflammatory properties. However, its role in modulating immune responses via the AhR signaling pathway remains unclear. In this study, 360 male Arbor Acre broilers (1-day-old) were fed a basal diet supplemented with 40 or 80 mg/kg genistein and infected with or without *Clostridium perfringens* (*Cp*). Our results demonstrated that genistein ameliorated *Cp*-induced intestinal damage, as reflected by the reduced intestinal lesion scores and improved intestinal morphology and feed-to-gain ratio. Moreover, genistein increased intestinal sIgA, TGF-β, and IL-10, along with elevated serum IgG, IgA, and lysozyme levels. Genistein improved intestinal AhR and cytochrome P450 family 1 subfamily A member 1 (CYP1A1) protein levels and AhR^+^ cell numbers in *Cp*-challenged broilers. The increased number of AhR^+^CD163^+^ cells in the jejunum suggested a potential association between genistein-induced AhR activation and anti-inflammatory effects mediated through M2 macrophage polarization. In IL-4-treated RAW264.7 cells, genistein increased the levels of AhR, CYP1A1, CD163, and arginase (Arg)-1 proteins, as well as IL-10 mRNA levels. This increase was attenuated by the AhR antagonist CH223191. In summary, genistein activated the AhR signaling pathway in M2 macrophages, which enhanced the secretion of anti-inflammatory cytokines and attenuated intestinal damage in *Cp*-infected broilers *Cp*.

## 1. Introduction

The aryl hydrocarbon receptor (AhR) is a ligand-activated transcription factor that integrates various ligands, such as environmental contaminants, dietary components, microbiota, and host metabolism [1]. Activation of the AhR induces the expression of cytochrome P450 enzymes (CYP1A1 and CYP1B1), which metabolize AhR ligands and terminate AhR activation [2]. The AhR, activated by various ligands, regulates intestinal immune homeostasis and protects from inflammation damage by maintaining the number of intraepithelial lymphocytes and innate lymphoid cells, as well as balancing T helper 17/regulatory T cells and M1/M2 macrophages [3,4,5,6]. AhR-deficiency contributes to the loss of these AhR-dependent intestinal immune cells, thereby increasing the susceptibility to enteral infection [4,7,8,9,10].

Genistein, the most active component of soy isoflavones, is a natural phytoestrogen with antioxidant, anti-inflammatory, and immunomodulatory properties [11,12,13]. Genistein is found to be an effective anti-inflammatory agent as demonstrated by inhibition of interferon regulatory factor-1 and phosphorylated STAT1, the transcription factor of inducible nitric oxide synthase, thereby reducing nitric oxide production induced by lipopolysaccharide (LPS) in the rat microglial cell line [14]. Moreover, genistein has been shown to attenuate intestinal inflammation through inhibition of the cyclooxygenase-2/prostaglandin E2, nuclear factor kappa-B, pro-inflammatory cytokines, and reactive oxygen species pathways [15,16,17,18]. In vivo studies have shown that dietary genistein supplementation has been observed to increase the number of intestinal intraepithelial lymphocytes and improve mucosal immunity in bacteria-challenged broilers [13,19]. Previous studies have shown that genistein is a weak AhR agonist [20,21,22], as indicated by the increased expression of AhR target genes such as CYP1A1 and CYP1B1 [23]. However, the precise role of the AhR pathway in the regulation of inflammatory responses by genistein has yet to be elucidated.

Necrotic enteritis (NE), mainly caused by *Clostridium perfringens* (*Cp*), has been a major economic problem for the poultry industry [24]. The toxins secreted by pathogenic strains result in epithelial cell damage, induce inflammatory responses, interrupt intestinal barrier function, and ultimately lead to the development of NE [25]. In the infection state, macrophages in the gut participate in the maintenance of intestinal immune homeostasis by regulating the balance of pro-inflammatory M1 and anti-inflammatory M2 phenotypes [26]. During the early infection stage, M1 macrophages promote inflammation by secreting TNF-α, IL-1β, IL-12, and IL-23, while M2 macrophages secrete IL-10 and TGF-β to inhibit inflammation and render tissue repair when tissue lesion occurs [26]. Previous research has found that AhR influences the dynamic balance between M1 and M2 macrophages, as proven by the overproduction of pro-inflammatory cytokines in AhR-null macrophages with LPS stimulation and by the lower levels of M2 macrophage markers in AhR-null macrophages with IL-4 stimulation [4,27].

Previous research has demonstrated genistein’s ability to mitigate gut inflammation [15]. Moreover, the AhR signaling pathway is known to modulate immune responses [5]. However, the potential for genistein, acting as a weak agonist of AhR, to influence the intestinal AhR signaling pathway and its target cells remains elusive. In this study, we constructed a necrotic enteritis model to elucidate the protective impact of genistein on intestinal inflammation. Our aim was to investigate the interplay between genistein and the AhR signaling pathway, including its target cells, thereby providing a clearer understanding of genistein’s protective effects on intestinal immune responses.

## 2. Results

### 2.1. Genistein Ameliorates Cp-Induced Intestinal Injury

In comparison with the control group, broilers challenged with *Cp* exhibited a significantly increased feed-to-gain (F/G) ratio during the 1–22 days, although no mortality was reported (Table 1). Conversely, supplementation of genistein in *Cp*-infection broilers decreased F/G compared with the *Cp* group (Table 1).

*Cp*-induced SNE caused severe intestinal erosion as evidenced by the separation of lamina propria and intestinal epithelium, necrosis, and exfoliation of numerous epithelial cells, which were alleviated by the addition of genistein (Figure 1A). Furthermore, the intestinal lesion scores were higher in the *Cp* group compared with the control group, which was alleviated by genistein supplementation (Figure 1B). Notably, only 40 mg/kg genistein supplementation significantly increased the duodenal villus height of broilers in comparison with the control group (Figure 1C). Additionally, *Cp* infection markedly reduced villus height and the ratio of villus height to crypt depth, whereas supplementation of genistein alleviated the negative impact of *Cp* on the villus height of the duodenum (Figure 1C).

### 2.2. Genistein Enhances Immune Responses

*Cp* infection significantly decreased the levels of mucosa sIgA, TGF-β, and IL-10 compared with the control group (Figure 2A). However, genistein supplementation significantly increased the levels of these immune factors, regardless of *Cp* infection status (Figure 2A). Notably, the addition of genistein led to a significant increase in the levels of IL-10 and sIgA compared with the control group (Figure 2A). Furthermore, the addition of genistein in the *Cp*-infected group markedly increased the levels of IgG, IgA, and lysozyme in the serum compared with the *Cp* group (Figure 2B).

### 2.3. Genistein Activates Intestinal AhR Signaling Pathway

To investigate the potential involvement of the AhR signaling pathway in intestinal mucosal immunity, we examined the abundance of AhR and its target gene, CYP1A1 proteins, in the jejunum of broilers with NE by western blotting. Our data demonstrated a significant decrease in the protein levels of AhR and CYP1A1 in *Cp*-infected broilers compared with the control group (Figure 3A,B). However, genistein addition significantly upregulated the abundance of AhR and CYP1A1 proteins in broilers with NE compared with the *Cp*-infected group (Figure 3A,B).

### 2.4. Genistein Promotes AhR Expression in Intestinal M2 Macrophages

Double-immunofluorescence (IF) staining was conducted to investigate the efficacy of immune cells in the intestinal mucosa. The IF analysis of AhR and CD163 demonstrated that the number of AhR^+^ and AhR^+^ CD163^+^ cells was not significantly affected by either genistein supplementation or *Cp* challenge compared with the control group. However, genistein supplementation markedly increased the number of AhR^+^ and AhR^+^ CD163^+^ cells in broilers challenged with NE in comparison with the *Cp* group (Figure 4A,B).

### 2.5. Genistein Induces M2 Macrophage Polarization via AhR

In an in vitro study, we investigated the influence of genistein on the polarization of M2 macrophages and its interrelation with AhR in RAW264.7 cells. Cell viability assay demonstrated that genistein at concentrations ranging from 25 to 200 μΜ did not affect cell viability (Figure 5A). Furthermore, we measured the mRNA levels of AhR and Arg-1 through qRT-PCR. We found that genistein at varying concentrations significantly increased the expression of AhR and Arg-1, with the most substantial effect observed at 50 μM in the presence of IL-4 stimulation in RAW264.7 cells, which was used for subsequent experiments (Figure 5B).

IL-4-stimulated macrophages were treated separately with genistein, FICZ (AhR agonist), CH223191 (AhR antagonist), or a combination of genistein and CH223191 in the presence of IL-4 for 24 h. The western blot analysis demonstrated that IL-4 treatment significantly induced the protein expression of AhR, CYP1A1, and CD163 in comparison with the control group. In comparison with the IL-4 stimulation group, the induction of AhR, CYP1A1, CD163, and Arg-1 proteins was significantly higher in the presence of FICZ and genistein, with FICZ demonstrating a more pronounced effect (Figure 5D). However, the above protein levels were suppressed by CH223191 treatment in the presence of genistein (Figure 5D). Notably, we observed similar changes in the mRNA levels of IL-10 (Figure 5C), which is the primary anti-inflammatory cytokine secreted by M2 macrophages.

## 3. Discussion

Subclinical necrotic enteritis (SNE), mainly caused by *Clostridium perfringens* (*Cp*), is characterized by poor growth performance, intestinal injury, and inflammatory responses [24]. Genistein has been shown to ameliorate intestinal injury caused by pathogenic bacteria and enhance mucosal immune function [16,17]. In this study, genistein ameliorated pathological alterations in the gut and poor growth performance induced by *Cp* infection, indicating a protective effect against intestinal injury. sIgA is a critical immunoglobulin present in the mucosal surface of the gut, which mediates innate and adaptive immune responses by resisting and coating bacteria [28,29]. TGF-β and IL-10, the main anti-inflammatory cytokines produced by M2 macrophages in the intestinal mucosa, are capable of suppressing the secretion of inflammatory cytokines and modulating immune responses [26,30,31]. Serum immunoglobulins and lysozyme play a crucial role in protecting the host from both internal and external threats [32,33]. Our results indicated that genistein enhanced immune function in broilers challenged with NE by increasing mucosal sIgA, TGF-β, and IL-10, as well as serum IgG, IgA, and lysozyme. Our recent study has shown that genistein improves intestinal mucosal barrier function and growth performance of *E. coli*-challenged broilers, evidenced by the elevation of sIgA and reduction of inflammatory cytokines [13]. Furthermore, the immunomodulatory properties of genistein are supported by an increase in broilers’ serum IgG and IgM concentrations and the restoration of IL-10 in the colon of mice with colitis [34,35].

Increasing evidence suggests that AhR, activated by both endogenous and exogenous ligands, plays a critical role in maintaining intestinal immune homeostasis by regulating AhR-dependent differentiation of immune cells and immune responses [36,37,38]. CYP1A1, the downstream target gene of AhR, is the indicator of the activation of AhR agonists, as previously demonstrated [2]. In vitro studies have demonstrated that genistein possesses a mild AhR agonistic effect, evidenced by its ability to induce CYP1A1 [23,39,40]. Our findings demonstrated that genistein has the potential to activate the AhR pathway, as evidenced by the increased abundance of AhR and CYP1A1 proteins, as well as an increase in AhR-positive cells in the intestine. Previous studies have shown that the ligand-activated AhR signaling pathway may be involved in regulating innate and adaptive immunity by directly or indirectly regulating the differentiation and function of dendritic cells, macrophages, and T cells [27,41,42]. A recent investigation revealed that the activation of AhR by the ligand FICZ mitigated intestinal inflammation through the modulation of the Th17/Treg balance [5]. Additionally, another study revealed that pectin suppressed inflammatory responses by reinforcing the AhR-IL22-STAT3 signaling pathway [43]. The results of our observations suggest a potential role for genistein in modulating intestinal mucosal immunity via the AhR/CYP1A1 signaling pathway.

Macrophages exert immunomodulatory effects on inflammatory responses by polarization into pro-inflammatory M1 or anti-inflammatory M2 phenotypes in response to stimulation by microenvironmental factors [26]. AhR is essential for the polarization and immune function of M2 macrophages. Activation of AhR has been shown to exhibit immunomodulatory effects by promoting the polarization of macrophages from pro-inflammatory M1 to anti-inflammatory M2 macrophages [44,45]. In this study, intestinal IF results for AhR and the M2 macrophage marker CD163 provided evidence that genistein promoted AhR expression in M2 macrophages of broilers with NE, consistent with variations in AhR protein levels. M2 macrophages typically secrete anti-inflammatory cytokines such as TGF-β and IL-10 to suppress inflammation [26]. Accordingly, increased levels of TGF-β and IL-10 induced by genistein supported the results that genistein-induced AhR contributed to the elevation of intestinal M2 macrophages. Additionally, our findings align with previous research showing that genistein mitigates colitis by shifting M1 macrophages toward the M2 phenotype and diminishing systemic cytokine levels [46].

In vitro, M2 macrophages are induced by IL-4 in RAW264.7 cells, leading to the upregulation of M2 markers such as CD163, Arg-1, IL-10, and Fizz-1 [47]. The in vitro study aimed to investigate the effect of genistein on M2 macrophage polarization in RAW264.7 cells through the AhR signaling pathway. Our study showed that genistein improved the abundance of AhR, CYP1A1, CD163, and Arg-1 proteins, as well as IL-10 mRNA levels in IL-4-treated cells. The effect of activated AhR on M2 macrophage differentiation was consistent with the previous finding [27], in which macrophage AhR promoted the expression of Arg-1 and IL-10. However, the induction effect of genistein was weakened in the presence of CH223191, an AhR antagonist, demonstrating that genistein promotes M2 macrophage polarization depending on the AhR signaling pathway. Previous research has also shown that the deletion of AhR in IL-4-treated M2 macrophages leads to reduced M2 markers, such as Fizz1, Ym1, and IL-10 [4], indicating the importance of AhR in the differentiation of M2 macrophages. Moreover, another investigation provided evidence that the flavonoid kurarinone regulated mucosal inflammation through macrophage-intrinsic AhR signaling [48]. Consequently, our investigation demonstrated that AhR, activated by genistein, plays a role in modulating intestinal M2 macrophages and suppressing inflammatory responses. In forthcoming research, we aim to further elucidate the molecular mechanisms underlying the functional role of macrophages upon AhR activation, thereby laying a theoretical foundation for its prospective clinical application.

## 4. Materials and Methods

### 4.1. Reagents

Dulbecco’s modified Eagle’s medium (DMEM), fetal bovine serum (FBS), and penicillin-streptomycin (PS, 10,000 U/mL) were purchased from Thermo Fisher Scientific Inc. (Waltham, MA, USA). Cell Counting Kit-8 (CCK-8) reagent was obtained from Yeasen (Shanghai, China). Dimethyl sulfoxide (DMSO) and 6-formylindolo[3,2-b]carbazole (FICZ) were purchased from Sigma-Aldrich (St. Louis, MO, USA). LPS, IL-4, genistein, and CH223191 were purchased from MedChemExpress (Monmouth Junction, NJ, USA). Trizol reagent, PrimeScriptTM RT Reagent Kit with gDNA Eraser, and TB Green^®^ Premix Ex Taq TM II were from Takara (Kusatsu, Japan). Antibodies used in Western blot and immunofluorescence included the following: rabbit anti-AhR (MBS829549, Mybiosoure, San Diego, CA, USA), goat anti-AhR antibody (NB100-128, Novus, Littleton, CO, USA), rabbit anti-CYP1A1 (D120518, Sangon biotech, Shanghai, China), rabbit anti-CD163 antibody (A8383, ABclonal, Wuhan, China), rabbit anti-Arginase-1(Arg-1, GB11285, Service-bio, Wuhan, China), mouse anti-β-actin (30101ES10, Yeasen, Shanghai, China), mouse anti-β-tubulin (M20005, Abmart, Shanghai, China), Alexa Fluor™ 488-conjugated anti-goat (A11055, ThermoFisher) and Alexa Fluor 647-conjugated anti-rabbit secondary antibody (A0468, Beyotime, Shanghai, China). The *Cp* type A strain was obtained from the China Veterinary Culture Collection Center (CVCC52, Beijing, China).

### 4.2. Animals

All animal procedures were performed in accordance with the Guidelines for Care and Use of Laboratory Animals, and all the experimental protocols for this study were approved by the Animal Care and Use Committee of China Agricultural University (no. AW12203202-1-2). A total of 360 1-day-old male Arbor Acre broilers were randomly assigned to 6 treatments, each consisting of 6 pens as replicates with 10 birds per pen. The treatments were as follows: Con group (a basal diet), Gen40 group (a basal diet supplemented with 40 mg/kg genistein), Gen80 group (a basal diet supplemented with 80 mg/kg genistein), *Cp* group (*Clostridium perfringens* infection from day 15 to 21), *Cp*+Gen40 group (Gen40 diet and *Cp* infection), *Cp*+Gen80 group (Gen80 diet and *Cp* infection). The diet provided to the birds from 1 d to 22 d was the wheat–corn–soybean meal basal diet that was formulated to meet or exceed the recommendations of the National Research Council (1994) of broilers. The composition and nutrient levels of the basal diet are shown in Table 2. Throughout the study, all birds had free access to feed and water. Genistein was purchased from Ruiying (Xi’an, China). The duration of the study was 22 days.

### 4.3. Necrotic Enteritis Model

The NE infection was based on the model originally developed by Dahiya et al. [49] with some modifications. The subclinical necrotic enteritis (SNE) model was established using the *Cp* type A strain. The *Cp* were anaerobically cultured in thioglycollate medium (02-015, Aoboxing Bio-tech Co., Ltd., Beijing, China) at 37 °C for 18 h and were quantified on tryptose–sulfite–cycloserine agar base (02-210, Aoboxing Bio-tech Co.) plates supplemented with D-cycloserine. From days 15 to 21, all birds underwent an 8-h fasting period before gavage. The *Cp*-infection birds were orally gavaged with 1mL (2 × 10^8^ CFU/mL) of *Cp*, while the uninfected birds were gavaged orally with 1 mL of sterile thioglycollate medium.

### 4.4. Intestinal Lesion Scores

The duodenum, jejunum, and ileum were isolated, and the intestinal lesions were scored by 3 individuals blinded to the trial. Lesions were scored on a 0–4 scale as described previously by Dahiya et al. [49], with 0: no visible lesions; 0.5: hyperemia of the mesentery and small intestinal wall; 1: thin and brittle small intestine with more than 5 small red petechiae; 2: gas in the intestinal lumen, with numerous bleeding spots on the intestinal wall, accompanied with few necrosis or ulcers; 3: regional necrosis or ulceration of the intestinal wall; 4: a large amount of gas in the gut lumen, severe and extensive mucosal necrosis.

### 4.5. Intestinal Morphology Analysis

The duodenum of broilers was fixed in the 4% paraformaldehyde solution, prepared into paraffin sections, and stained with hematoxylin-eosin (HE). The integrity of the intestinal villi was assessed using a Nikon Eclipse E200 microscope, with images captured for documentation. Ten intact intestinal villi were randomly selected from each section to measure the villus height (VH) and crypt depth (CD) under microscopic observation. Additionally, the ratio of villus height to crypt depth (VH/CD) was calculated to provide a comprehensive evaluation of the intestinal morphology.

### 4.6. Serum and Mucosal Immune Parameters

The quantitative determination of immunoglobulin G (IgG) and IgA concentration in serum and secretory IgA (sIgA), TGF-β, and IL-10 in jejunal mucosa was carried out using enzyme-linked immunosorbent assay (ELISA) kits according to the instructions (Enzyme-linked Biotech Co., Shanghai, China). Lysozyme activity was measured using lysozyme assay kits (Jiancheng Bioengineering Institute, Nanjing, China). The results are presented as the content of sIgA and cytokines per milligram of protein in the jejunum of broilers.

### 4.7. Western Blot Analysis

The total protein was extracted with RIPA buffer (P0013B, Beyotime) containing protease inhibitor and phosphatase inhibitor cocktail (P1048, Beyotime). The supernatants were collected after centrifugation at 12,000 rpm for 10 min, and the total protein concentrations were determined using the PierceTM BCA protein assay kit (23227, Thermo Fisher Scientific). The proteins were separated by 10% SDS-PAGE and transferred onto PVDF membranes (IPVH00010, Millipore, Burlington, MA, USA). The membranes were blocked with 5% BSA for 1 h at room temperature, followed by overnight incubation at 4 °C with primary antibodies, including rabbit anti-AhR (1:1000), rabbit anti-CYP1A1 (1:200), and mouse anti-β-actin (1:10,000). After washing with TBST buffer, the membranes were incubated with anti-rabbit (1:5000) or anti-mouse secondary antibody (1:5000). The protein signals were visualized using enhanced chemiluminescence substrate (P0018S, Beyotime) and quantified using Fiji Image J software (https://imagej.net/software/fiji/ accessed on 10 March 2024).

### 4.8. Immunofluorescence

The jejunal tissue was fixed in 4% paraformaldehyde solution, embedded in paraffin, and sectioned into 4 μm slices. The slices underwent a series of incubation steps in different ethanol concentrations and were then retrieved with a citric acid antigen repair solution. The slices were blocked with 10% sheep serum for 1 h and incubated with goat anti-AhR antibody (1:100) and rabbit anti-CD163 antibody (1:100) at 4 °C overnight. Slices were incubated with Alexa Fluor™ 488-conjugated anti-goat (1:500) and Alexa Fluor 647-conjugated anti-rabbit secondary antibody (1:500) at room temperature for 1 h. The slices were then incubated with DAPI to stain the nucleus and immediately photographed using a confocal microscope (Nikon AX, Tokyo, Japan).

### 4.9. In Vitro Experiment

#### 4.9.1. Cell Culture and Treatment

RAW264.7 cells, a murine macrophage cell line, were obtained from Professor Wu of the China Agricultural University (Beijing, China). These cells were cultured in DMEM containing 10% (*v*/*v*) FBS, 100 U/mL PS at 37 °C with 5% CO_2_. To assess the impact of genistein on the viability of RAW264.7 cells, cells were seeded overnight at a density of 1 × 10^4^ cells/well in 96-well plates and then exposed to genistein (0, 25, 50, 100, 200 μM) for 48 h, followed by the addition of CCK-8 reagent and incubation for 1 h at 37 °C. The absorbance was measured at 450 nm with a Microplate Reader (Thermo Fisher Scientific). RAW264.7 cells were polarized into M2 macrophages with IL-4 (20 ng/mL) treatment for 24 h. Cells were then treated with DMSO (0.1%), genistein (50 μM), AhR agonist FICZ (100 nM), and AhR antagonist CH223191 (5 μM) with or without genistein in the presence of IL-4 for 24 h.

#### 4.9.2. Quantitative Real-Time PCR

The total RAN of cells was extracted with Trizol reagent, and its concentration was determined at 260 and 280 nm using a Nano-300 micro-spectrophotometer (Allsheng, Hangzhou, China). RNA was reversely transcribed into cDNA utilizing the Prime-ScriptTM RT Reagent Kit with gDNA Eraser. Quantitative real-time PCR (qRT-PCR) was performed with the TB Green^®^ Premix Ex Taq TM II using the 7500 Real-Time PCR System (Applied Biosystems, Thermo Fisher Scientific). β-actin was used as the internal control gene. The primer sequences are provided in Table 3. The qPCR amplification was performed as follows: 95 °C for 30 s and then 40 cycles of 5 s at 95 °C and 34 s at 60 °C. The relative mRNA expression levels were calculated by using the 2^−ΔΔCT^ method.

### 4.10. Statistical Analysis

All results were analyzed by one-way ANOVA followed by Duncan’s test using SPSS 25.0 software (SPSS for Windows, version 25.0, Chicago, IL, USA). Results are presented as mean ± SEM and statistical significance was set at *p* < 0.05.

## 5. Conclusions

In summary, our results suggest that genistein ameliorates intestinal damage and improves growth performance in broilers with necrotic enteritis through its immunomodulatory effects. Specifically, genistein suppresses intestinal inflammatory responses by promoting M2 macrophage polarization and inducing the secretion of anti-inflammatory cytokines in an AhR-dependent manner.

## Figures and Tables

**Figure 1 ijms-25-06656-f001:**
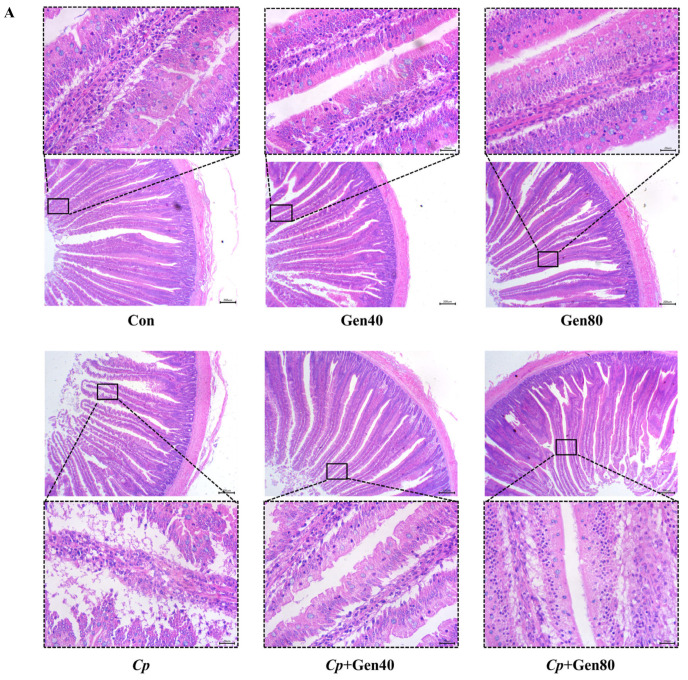
Genistein alleviates intestinal damage in broilers infected with *Clostridium perfringens*. (**A**) Hematoxylin and eosin (HE) staining of duodenum of broilers (×40 magnification, scale bar = 200 μm). (**B**) The intestinal lesion scores of broilers. (**C**) The villus height (VH), crypt depth (CD), and the ratio of VH to CD (VH/CD) in the duodenum of broilers. Con, basal diet; Gen40, basal diet supplemented with 40 mg/kg genistein; Gen80, Gen80 diet; *Cp*, basal diet and *Cp* infection; *Cp*+Gen40, Gen40 diet and *Cp* infection; *Cp*+Gen80, Gen80 diet and *Cp* infection. The differences among groups were determined by ANOVA using Duncan’s test. The results are presented as mean ± SEM, *n* = 12. Letter a indicates *p* < 0.05 vs. Con group; letter b indicates *p* < 0.05 vs. *Cp* group.

**Figure 2 ijms-25-06656-f002:**
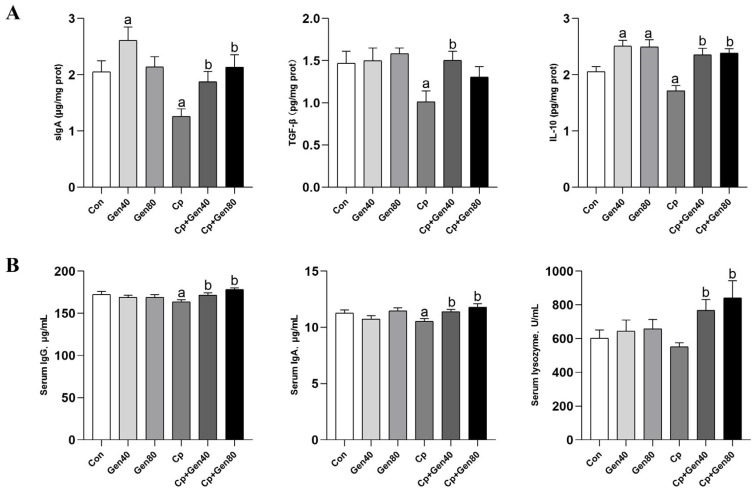
Genistein enhances immune responses in the intestine and serum of broilers. (**A**) The content of sIgA, TGF-β, and IL-10 in the jejunal mucosa of broilers was measured by ELISA kits. (**B**) The concentration of serum IgG and IgA of broilers was determined by ELISA kits. The lysozyme activity was measured by lysozyme assay kits. Con, basal diet; Gen40, basal diet supplemented with 40 mg/kg genistein; Gen80, Gen80 diet; *Cp*, basal diet and *Cp* infection; *Cp*+Gen40, Gen40 diet and *Cp* infection; *Cp*+Gen80, Gen80 diet and *Cp* infection. The differences among groups were determined by ANOVA using Duncan’s test. The results are presented as mean ± SEM, *n* = 12. Letter a indicates *p* < 0.05 vs. Con group; letter b indicates *p* < 0.05 vs. *Cp* group.

**Figure 3 ijms-25-06656-f003:**
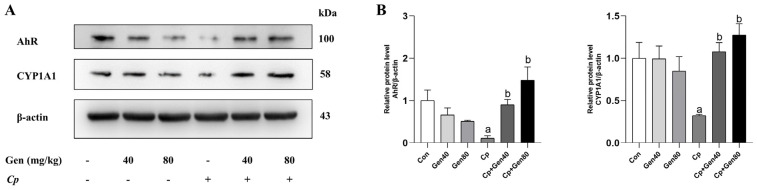
Genistein activates the intestinal AhR pathway in broilers challenged with NE. (**A**) Western blot of AhR and CYP1A1 in the jejunum of broilers. (**B**) Quantification of AhR and CYP1A1 protein levels, *n* = 3. Con, basal diet; Gen40, basal diet supplemented with 40 mg/kg genistein; Gen80, Gen80 diet; *Cp*, basal diet and *Cp* infection; *Cp*+Gen40, Gen40 diet and *Cp* infection; *Cp*+Gen80, Gen80 diet and *Cp* infection. The differences among groups were determined by ANOVA using Duncan’s test. The results are presented as mean ± SEM. Letter a indicates *p* < 0.05 vs. Con group; letter b indicates *p* < 0.05 vs. *Cp* group.

**Figure 4 ijms-25-06656-f004:**
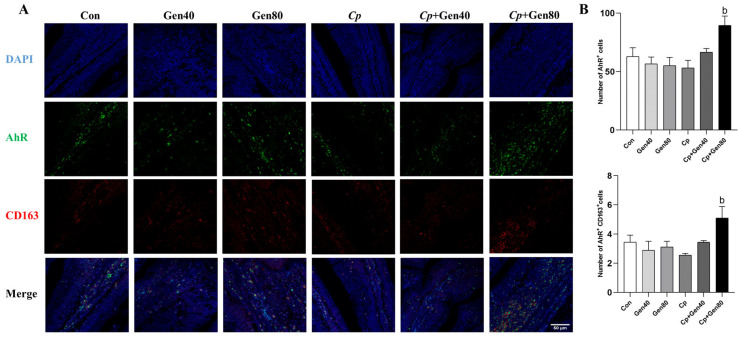
Genistein promotes AhR expression in intestinal M2 macrophages of broilers. (**A**) Sections of jejunum tissues were immunostained with DAPI (blue), AhR (green), and CD163 (red) (×400 magnification, scale bar = 50 μm). (**B**) The quantified number of AhR^+^ and AhR^+^CD163^+^ cells in the jejunum of broilers, *n* = 3. Con, basal diet; Gen40, basal diet supplemented with 40 mg/kg genistein; Gen80, Gen80 diet; *Cp*, basal diet and *Cp* infection; *Cp*+Gen40, Gen40 diet and *Cp* infection; *Cp*+Gen80, Gen80 diet and *Cp* infection. The differences among groups were determined by ANOVA using Duncan’s test. The results are presented as mean ± SEM. Letter b indicates *p* < 0.05 vs. *Cp* group.

**Figure 5 ijms-25-06656-f005:**
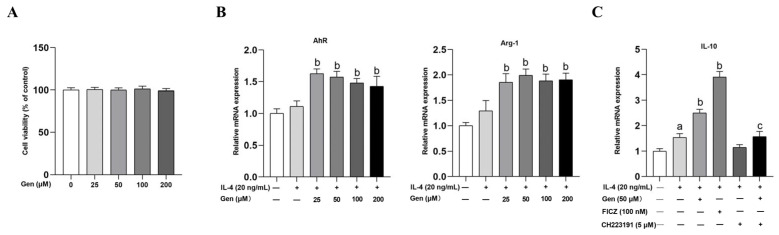
Genistein promotes M2 macrophage polarization depending on AhR in RAW264.7 cells. (**A**) Cells were treated with genistein (0, 25, 50, 100, 200 μM) for 48 h. The effect of genistein on cell viability was determined by the CCK-8 assay, *n* = 10. (**B**) Cells were prestimulated with IL-4 (20 ng/mL) for 24 h and then treated with genistein (0, 25, 50, 100, 200 μM) and IL-4 (20 ng/mL) for 24 h. The relative mRNA expression levels of AhR and Arg-1 were measured by qRT-PCR. (**C**) Cells were prestimulated with IL-4 (20 ng/mL) for 24 h and then treated with genistein (50 μM), FICZ (100 nM), CH223191 (5 μM), genistein + CH223191 in the presence of IL-4 for 24 h. The relative mRNA expression levels of IL-10 were measured by qRT-PCR. (**D**) The protein levels of AhR, CYP1A1, CD163, and Arg-1 were determined by Western blot. The differences among groups were determined by ANOVA using Duncan’s test. The results are presented as mean ± SEM, *n* = 3. Letter a indicates *p* < 0.05 vs. Con group; letter b indicates *p* < 0.05 vs. IL-4-treatment group; letter c indicates *p* < 0.05 vs. Gen-treatment group.

**Table 1 ijms-25-06656-t001:** Effect of genistein on the growth performance of broilers. Con, basal diet; Gen40, basal diet supplemented with 40 mg/kg genistein; Gen80, Gen80 diet; *Cp*, basal diet and *Cp* infection; *Cp*+Gen40, Gen40 diet and *Cp* infection; *Cp*+Gen80, Gen80 diet and *Cp* infection. The differences among groups were determined by ANOVA using Duncan’s test. The results are presented as mean ± SEM, *n* = 6. Letter a indicates *p* < 0.05 vs. Con group; letter b indicates *p* < 0.05 vs. *Cp* group.

Items	Con	Gen40	Gen80	*Cp*	*Cp*+Gen40	*Cp*+Gen80	*p*-Values
Day 22							
BW, g	902 ± 15.6	928 ± 18.2	890 ± 18.9	884 ± 3.2	909 ± 17.1	905 ± 9.9	0.391
Days 1–22							
ADG, g/d	39.1 ± 0.71	40.3 ± 0.82	38.5 ± 0.86	38.3 ± 0.14	39.4 ± 0.78	39.2 ± 0.45	0.384
ADFI, g/d	57.9 ± 0.88	58.7 ± 1.93	57.0 ± 0.95	59.4 ± 1.18	58.1 ± 0.48	59.6 ± 0.72	0.575
F/G	1.48 ± 0.010	1.45 ± 0.024	1.48 ± 0.015	1.55 ± 0.027 ^a^	1.48 ± 0.02 ^b^	1.52 ± 0.017 ^b^	0.025

**Table 2 ijms-25-06656-t002:** Composition and nutrient levels of the basal diet (air-dry basis).

Ingredients (%)	Days 1–22	Calculated Nutrient Levels	Days 1–22
Corn	44.15	ME (Mcal/kg)	2.95
Soybean meal	25.65	Crude protein (%)	20.50
Wheat	20.00	Lysine (%)	1.15
Corn gluten meal	3.59	Methionine (%)	0.50
Soybean oil	2.00	Calcium (%)	1.01
Calcium hydrogen phosphate	2.00	Available phosphorus (%)	0.47
Limestone	1.20		
Sodium chloride	0.35		
*L*-Lysine HCl (78%)	0.40		
Choline chloride	0.20		
Trace mineral premix ^1^	0.20		
*DL*-Methionine	0.18		
Vitamin premix ^2^	0.02		
Zeolite powder	0.06		
Total	100		

^1^ Trace mineral premix (per kg of diet): Mn, 100 mg; Zn, 50 mg; Fe, 60 mg; Cu, 6 mg; I, 0.35 mg; Se, 0.15 mg. ^2^ Vitamin premix (per kg of diet): vitamin A, 10,800 IU; vitamin D3, 2160 IU; vitamin E, 4.6 mg; vitamin K3, 1.0 mg; vitamin B1, 0.4 mg; vitamin B2, 5 mg; vitamin B12, 6 mg; folic acid, 0.1 mg; pantothenic acid 5 mg; niacin, 7 mg.

**Table 3 ijms-25-06656-t003:** Primer pairs for qRT-PCR analysis.

Genes	Primer Sequences (5′-3′)
β-actin	F: CTTCTTTGCAGCTCCTTCGTT
	R: AGGAGTCCTTCTGACCCATTC
AhR	F: CTTAGGCTCAGCGTCAGTTAC
	R: CGTTTCTTTCAGTAGGGGAGGAT
Arg-1	F: CTCCAAGCCAAAGTCCTTAGAG
	R: AGGAGCTGTCATTAGGGACATC
IL-10	F: GCTCTTACTGACTGGCATGAG
	R: CGCAGCTCTAGGAGCATGTG

## Data Availability

All data are contained within the article.

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
