# Peer review of "Genistein Promotes M2 Macrophage Polarization via Aryl Hydrocarbon Receptor and Alleviates Intestinal Inflammation in Broilers with Necrotic Enteritis"

_ijms, 2024, doi:10.3390/ijms25126656_

Round 1
Reviewer 1 Report
Comments and Suggestions for Authors
This study investigated the impact of genistein on intestinal immune function through the AhR signaling pathway. The experimental design was meticulously executed, and the model of necrotic enteritis in broilers was successfully established. The utilization of immunofluorescence double staining technique effectively examined the influence of genistein on M2 macrophages, the target cells in the intestine. Furthermore, in vitro experiments using mouse macrophage cell lines were conducted to validate the findings from animal experiments. The exploration of genistein dosage and experimental treatments was well justified. The setting of genistein dose exploration and experimental treatments were reasonable. In comparison with previous studies, this research successfully demonstrated the involvement of the AhR signaling pathway in modulating the secretion of anti-inflammatory cytokines by M2 macrophages.
I appreciate the overall quality of the manuscript, which portrayed a comprehensive and well-structured analysis. The content followed the journal's guidelines, suggesting a careful preparation by the authors. However, I did notice a few grammatical errors, which I have pointed out in my minor comments. Additionally, some important points lacked sufficient description. Therefore, I recommend a minor revision before considering it for publication. The comments provided below must be addressed during the revision process.
Major comments:
1. In the introduction section, the authors need to provide detailed information on current progress in genistein as a weak AhR agonist.
2. In the discussion, please use appropriate connective words to improve the logical connection between sentences and the fluency of the text.
Minor comments:
1. Page 1, lines 16-20. Please provide full names of CYP1A1 and Arg-1 in the abstract.
2. Page 2, lines 58-62. Consider using “research” rather than “researches” and modifying the predicate. Please remove the dash between IL-4 and stimulation.
3. Page 2, line 65 and Page 7, line 188. Suggest to change the “immune response” to “immune responses” and keep the whole article consistent.
4. Page 2, lines 63-67. The research objective and conclusion are not clear, please revise them and clarify the mechanism.
5. Page 4, line 116. Please remove the dash between “mu” and “cosal”.
6. Page 5, line 146. Italics are recommended for "in vitro".
7. Page 7, lines 209-211. I think the speculation is insufficient, please add references to clarify the relationship between the previous studies and yours.
8. Page 7, lines 237-238. There is no concluding statement at the end of the paragraph, please supplement the conclusion by combining the results of previous research and this experiment.
Comments on the Quality of English LanguageThe quality of English is good and can be further improved by some minor corrections.
Reviewer 2 Report
Comments and Suggestions for Authors
The authors presented a well designed and executed experiment that investigated the role of Genistein in alleviating NE in broilers by regulating the AhR and M2 macrophage. The paper is overall well written, only some minor suggestions are as follows:
L63-64, The authors should provide the objective and current knowledge gap here in the introduction instead of providing the conclusion of the study.
L100, sig-nifi-cantly remove the first "-"; L135 cha llenged, remove the space.
L275, Were the birds fasted for 8 hours and gavaged everyday from day 15-21? Do the authors have any references for this 7-day-consective challenge model, since to the reviewer's knowledge, usually NE infection was induced by 3-5 day inoculation.
L278, Did the authors come up with the Lesion scoring system? If not, please provide the reference. Please provide the information of the microscope used if the authors performed a microscopic lesion scoring.
L290, How was the samples further analyzed? Observed under microscope and measured with certain software? Please be specific.
L305, Please provide the manufacturer information for them even though these might be basic reagents.
L350, Did the authors use the student's t test to separate the means (comparing the results among each treatment)? If yes, the authors have to change to the correct way to do this which is the Tukey HSD or Duncan's test as these are the correct post hoc test, Using the student t test for multiple comparison significantly increase Type I error.
L349, The following is just the suggestion of the reviewer that the authors could consider using the more appropriate way to perform the statistical analysis. Since the experimental design was a 2x3 factorial arrangement (2: NE challenge or not x 3: three levels of Genistein), the best way to analysis the results would be a two-way ANOVA instead of one-way. This way the authors could see whether the NE infection interact with the Genistein supplementation as well as the main effects of the two factors. However, the reviewer understand that by doing so will provide too much workload for the authors as they might need to reproduce the figures and the results session, and the current version of the manuscript already provide enough information to support their hypothesis and conclusion. Thus, this is just a suggestion and the authors could use this for their future references.
